# SEQUENCE-LEVEL CERTAINTY REDUCES HALLUCINATION IN KNOWLEDGE-GROUNDED DIALOGUE GENERATION

## ABSTRACT

Model hallucination has been a crucial interest of research in Natural Language Generation (NLG). In this work, we propose sequence-level certainty as a common theme over hallucination in NLG, and explore the correlation between sequence-level certainty and the level of hallucination in model responses. We categorize sequence-level certainty into two aspects: probabilistic certainty and semantic certainty, and reveal through experiments on Knowledge-Grounded Dialogue Generation (KGDG) task that both a higher level of probabilistic certainty and a higher level of semantic certainty in model responses are significantly correlated with a lower level of hallucination. What's more, we provide theoretical proof and analysis to show that semantic certainty is a good estimator of probabilistic certainty, and therefore has the potential as an alternative to probability-based certainty estimation in black-box scenarios. Based on the observation on the relationship between certainty and hallucination, we further propose Certainty-based Response Ranking (CRR), a decoding-time method for mitigating hallucination in NLG. Based on our categorization of sequence-level certainty, we propose 2 types of CRR approach: Probabilistic CRR (P-CRR) and Semantic CRR (S-CRR). P-CRR ranks individually sampled model responses using their arithmetic mean log-probability of the entire sequence. S-CRR approaches certainty estimation from meaning-space, and ranks a number of model response candidates based on their semantic certainty level, which is estimated by the entailment-based Agreement Score (AS). Through extensive experiments across 3 KGDG datasets, 3 decoding methods, and on 4 different models, we validate the effectiveness of our 2 proposed CRR methods to reduce model hallucination.

## 1 INTRODUCTION

Previous works have researched the problem of hallucination in Natural Language Generation (NLG) (Ji et al., 2023; Lee et al., 2018; Filippova, 2020; Zhou et al., 2021; Maynez et al., 2020; Huang et al., 2021; Santhanam et al., 2021; Honovich et al., 2022; Pagnoni et al., 2021). Specifically, a number of prior studies have explored hallucination issues in the Knowledge-Grounded Dialogue Generation (KGDG) task (Li et al., 2019; Shuster et al., 2021; Santhanam et al., 2021; Honovich et al., 2021; Dziri et al., 2022b; Rashkin et al., 2021), in which a dialogue model is provided with textual knowledge and a series of conversation histories, and is expected to generate informative and meaningful responses to the previous conversation with the provided knowledge (Li et al., 2022). Under KGDG's task definition, a model response is said to be "hallucinated" if it is inconsistent or unsupported by the provided knowledge in the model input (Filippova, 2020; Dziri et al., 2022a).

Dialogue models still struggle to generate faithful responses on the KGDG task (Honovich et al., 2021). Understanding and mitigating hallucination in KGDG models therefore remains an important research question in the domain of NLG. Among previous works, Xiao & Wang investigated model hallucination on Image Captioning (IC) task. They proposed to use token-level probabilistic entropy to measure predictive uncertainty, and were the first to show that higher predictive uncertainty is associated with higher chance of hallucination. However, the proposed probability-based estimation of model uncertainty in their study remains on token-level, and therefore neglects sequence-level semantic information that might also indicate uncertainty in model generations. Token-level

probability-based predictive uncertainty is therefore not a satisfactory and generalizable common theme across model hallucinations in NLG.

Our work proposes and investigates **sequence-level certainty** as a better and more general common theme over hallucination phenomenon in language generation. Specifically, we dissect sequence-level model certainty into two categories: **probabilistic certainty** and **semantic certainty**. Through comprehensive experiments across 4 models on KGDG task, we empirically show that both a higher level of probabilistic certainty and a higher level of semantic certainty are positively and significantly correlated with a lower level of hallucination. In addition, we demonstrate through statistical proof that semantic certainty is a good estimator to probabilistic certainty. This proves semantic certainty as a promising alternative to probabilistic certainty in black-box scenarios, where model output probabilities are not always available.

Furthermore, based on our observation of the correlation between both probabilistic and semantic certainty and level of hallucination in model generations, we propose Certainty-based Response Ranking (CRR) to mitigate hallucination of KGDG models during decoding time. Specifically, aligning with our categorization of sequence-level certainty, we establish 2 types of CRR approaches: Probabilistic CRR (P-CRR), and Semantic CRR (S-CRR). P-CRR simply ranks a number of independently sampled model responses by their probabilistic certainty, which we follow previous work to establish as the arithmetic mean log-probability over entire sequences Kuhn et al. (2023); Murray & Chiang (2018). Since we observed that higher probabilistic certainty corresponds to lower hallucination level, P-CRR can effectively mitigate model hallucination during decoding time. S-CRR approaches certainty estimation from a semantic perspective, and ranks various independently sampled model response candidates by their semantic certainty. To measure semantic certainty, our study proposes to utilizes Agreement Score (AS), which is defined as the overall level of semantic entailment of each candidate with all other candidates. Since the calculation of AS takes into account the level of semantic alignment between candidate responses, AS is a good proxy to measure semantic certainty level in model generation, further allowing the ranking mechanism of S-CRR to finally select the output candidate with highest level of semantic certainty. Since higher semantic certainty is correlated with lower level of hallucination, S-CRR can also effectively reduce hallucination in model responses.

We validate the effectiveness of our P-CRR and S-CRR methods through extensive empirical analysis. Specifically, we conduct comprehensive experiments on 3 KGDG datasets, 4 NLG models with varied sizes, with 3 different decoding methods. Strong experiment results demonstrate that both P-CRR and S-CRR significantly reduce hallucinations in model outputs across all experiment settings. conduct ablation experiments to further investigate how the number of response candidates sampled for ranking can influence performance of CRR methods. Interestingly, we observe that S-CRR achieves significant improvement with increased number of response candidates, whereas P-CRR maintains similar performance. This indicates that S-CRR better captures semantic certainty in model generations through more pair-wise alignment analysis between additional response candidates, which further proves the effectiveness of the S-CRR method. Our work provides novel and significant findings on the relationship between sequence-level certainty and hallucination, therefore opening up a new direction for future researches to further explore and understand hallucination phenomenon in NLG.

## 2 BACKGROUND ON UNCERTAINTY AND HALLUCINATION

Previous research has explored and proposed different methods to measure uncertainty in model predictions (Xiao & Wang, 2021; Kuhn et al., 2023; Manakul et al., 2023). Among these works, Xiao & Wang was the first to explore the relationship between token-level predictive uncertainty and hallucination on the Image Captioning (IC) task. However, their proposed uncertainty estimation did not consider sequence-level or semantic-level information in generated text. Kuhn et al. further extended the estimation of model uncertainty to semantic space on the QA task, but did not explore how semantic uncertainty measurement is related to level of hallucination. Our study draws inspirations from these works, and is among the first to comprehensively and systematically explore the relationship between both sequence-level probabilistic and semantic certainty and level of hallucination in generative models. In this section, we briefly introduce the methods proposed by previous works to estimate uncertainty, as well as provide the background on the correlation between uncertainty and hallucination, as observed by prior studies.

## 2.1 Uncertainty Estimation in Generative Models

We define and categorize uncertainty estimation in NLG into two aspects: **probabilistic uncertainty** and **semantic uncertainty**. Xiao & Wang was the first to propose token-level predictive uncertainty, which quantifies the entropy of the token probability distribution that a model predicts in language generation. Their work adopts a probability-based approach and formulates the total predictive uncertainty of a predicted token as its entropy. We denote this method as the **probabilistic uncertainty** estimation. Kuhn et al.'s work, on the other hand, extends the exploration of uncertainty from the probabilistic perspective to the semantic aspect. They propose **semantic uncertainty** to examine uncertainty in meaning-space, and establish it as the entropy of the random variable representing the output distribution in the semantic event-space. Specifically, their work cluster sampled sequences of a language model given the same sequences into semantic equivalent classes, and formulate semantic uncertainty as the entropy of predicting a specific semantic equivalent class.

## 2.2 On Uncertainty and Hallucination

Xiao & Wang's work was the first to explore the correlation between model uncertainty and hallucination. Through experiments on IC task, they observed that higher token-level predictive uncertainty corresponds to a higher chance of hallucination. Based on this observation, they further proposed uncertainty-aware beam search, which takes into account the predictive uncertainty during beam search to reduce hallucination in model-predicted captions for images. Manakul et al.'s work applied the probabilistic uncertainty measurement to QA task, and showed that probability-based model uncertainty can be used to detect hallucinations in LLM responses; no additional insights on the relationship between model uncertainty and hallucination were provided in this study.

## 2.3 Limitations and Challenges

Despite previous efforts, current explorations on the correlation between model uncertainty and hallucination still suffer from a number of limitations and challenges. Although Xiao & Wang's observation on the relationship between uncertainty and hallucination is promising, their experiment and analysis only explored IC task. In addition, their proposed uncertainty-aware beam search method for hallucination mitigation cannot be extended for application on other decoding methods in language generation, such as nucleus sampling and top-k sampling. Furthermore, their token-level probability-based uncertainty estimation approach fails to take into account the sequence-level or semantic component in generated texts. Kuhn et al.'s work extended the concept of uncertainty estimation in language generation from the semantic aspect. However, their study only explored how semantic uncertainty is more predictive of model accuracy on Question Answering (QA) datasets, and did not explore the relationship between semantic uncertainty and level of hallucination. In addition, since the scope of their experiments is limited to the QA task, each input question in their experiments always has one corresponding ground truth answer. This means that their observation cannot be extended to more general language generation tasks, such as dialogue generation, which might not have specific "correct answer"s. The relationship of semantic uncertainty and hallucination remains unexplored. Therefore, given the limitations of previous works, establishing more generalizable uncertainty-related common themes over hallucination phenomenon in more general NLG settings remains a significant challenge.

## 3 Sequence-Level Certainty

Previous work showed that a higher level of token-level probabilistic predictive uncertainty in model generation is associated with a higher chance of hallucination (Xiao & Wang, 2021). Inspired by prior study, we propose **sequence-level model certainty** as a more general common theme across hallucination phenomena in a broader sense of language generation task. We further dissect sequence-level certainty into **probabilistic certainty** and **semantic certainty**. In this section, we first provide a definition of the 2 types of certainty in our study. Through experiments on multiple models and datasets on KGDG task, we show that **both a higher level of probabilistic certainty and a higher level of semantic certainty are positively correlated with a lower level of hallucination in model responses.** What's more, we provide a rigorous proof on how semantic certainty is a good estimator to probabilistic certainty, and therefore is a promising alternative to probabilistic

certainty when applied to black-box LLMs, for which probability of model outputs is not always available.

## 3.1 DEFINITION

**Probabilistic Certainty** Following previous works (Kuhn et al., 2023; Murray & Chiang, 2018), we define sequence-level probabilistic certainty of a generated sequence to simply be the arithmetic mean log-probability of the entire sequence. Given a generated sequence $s$ with length $N$, the sequence-level probabilistic certainty can be calculated as: $\frac{1}{N} \sum_{i=1}^{N} \log p(s_i|s_{<i})$, where $p(s_i|s_{<i})$ is the conditional probability of generating token $s_i$ in sequence $s$ given past tokens. This stands in contrast with the previously proposed probabilistic uncertainty measures, which only takes into account token-level information.

**Semantic Certainty** We define sequence-level semantic certainty to be the level of confidence that a model generates semantic contents of a response. For instance, suppose that we individually sample $n$ model outputs given the same context, among which $80\%$ consists of semantic contents that aligns with each other. Give the high confidence of the model to produce these responses with aligned semantic contents, we can say that these outputs have **high semantic certainty** level. In contrast, if semantic contents of the rest of the $20\%$ model outputs don't align with any other responses, then we can say that these outputs have **low semantic certainty** level.

## 3.2 USING AGREEMENT SCORE TO ESTIMATE SEMANTIC CERTAINTY

In order to quantify the estimation of semantic certainty in language generation, we propose to use Agreement Score (AS) as a proxy of semantic certainty. We hereby provide the definition of AS. Given a context $x$, we individually sample $N$ model response candidates to constitute set $\mathbb{S} = \{s^{(1)}, s^{(2)}, ..., s^{(n)}\}$. Let the relation Entailment$(\cdot, \cdot)$ denote the probability that two generated sequences entail each other, or semantically support each other. Then, the AS of model response $s^{(i)}$ can be calculated as:

$$AS(s^{(i)}) = \sum_{j=1}^{n} \text{Entailment}(s^{(i)}, s^{(j)})$$

(1)

, which is the summed probability of semantic entailment between $s^{(i)}$ and all other candidates.

## 3.3 EXPERIMENTS

We design and conduct experiments on the KGDG task to further explore the correlation between sequence-level certainty and level of hallucination. KGDG task provides a model with a series of dialogue history and a piece of textual knowledge, and requires the model to generate a response to the dialogue history according to the provided knowledge. Therefore, responses generated by a faithful KGDG model should be truthful to the knowledge provided in its input. We follow previous work (Dziri et al., 2022a)'s method to train $4$ KGDG models with different structures and sizes, and then explore the correlation between the $2$ types of sequence-level certainty and the level of hallucination in model response candidates. Experiment details are provided in this section.

### 3.3.1 IMPLEMENTATION

**Model Selection** For the KGDG task, we select 4 different base models of different sizes and structures: GPT2-small, GPT2-medium (Radford et al., 2019), T5-base (Raffel et al., 2020), and OpenLlama (Geng & Liu, 2023). For calculating the pairwise entailment probability during agreement score calculation, we utilize an off-the-shelf RoBERTa-Large-based (Liu et al., 2019) Natural Language Inference (NLI) model (Nie et al., 2020) and utilize the probability of entailment between pairs of response candidates. For hallucination evaluation, we use an off-the-shelf RoBERTa-Large-based hallucination classification model trained on FaithCritic, which is a derivative of the FaithDial dataset (Dziri et al., 2022a). Given a piece of knowledge and a model response, the hallucination classification model classifies the model's output as either being "hallucinated" or "faithful".

**Dataset** For training the KGDG model, we utilize FaithDial (Dziri et al., 2022a), a faithful knowledge-grounded dialogue corpus built from the Wizard of Wikipedia dataset (Dinan et al.,

2019). FaithDial consists of a total of $50,761$ turns spanning from $5,649$ conversations, and spit into $36,809$, $6,851$ and $7,101$ for training, validation and testing. We utilize the full training, validation, and test sets of FaithDial for training, selecting, and evaluating the KGDG model.

**Training Details**   Following hyper-parameter settings in Dziri et al. (2022a), we train the KGDG model for 10 epochs with batch size set to 16 and maximum sequence length set to 512. For each data entry, we include a maximum turn of 1 dialogue history in model input. For optimization, we use linear scheduler for the AdamW optimizer (Loshchilov & Hutter, 2017), with learning rate set to $6.25 \times 10^{-5}$, warmup ratio set to $0.04$, epsilon set to $1 \times 10^{-8}$, and weight decay set to 0. Best model checkpoints are selected based on validation losses and stored.

**Inference Details**   During inference time, we select nucleus + top-k sampling decoding method for generation. We set temperature to $1.0$, top k to 50, top p to $0.9$, and maximum new tokens to 100. All hyper-parameters for generation are selected to ensure best possible quality of generated text.

### 3.3.2   REPORTED METRICS

In order to investigate the correlation between sequence-level certainty and level of hallucination on KGDG task, we wish to conduct experiments to prove 2 hypotheses:

1. Faithful model responses have higher certainty than hallucinated answers.
2. A higher certainty is positively and significantly correlated with a lower level of hallucination.

To prove Hypothesis 1, for each input in FaithDial's test dataset, we individually sample 5 candidate responses from a KGDG model. Then, we calculate the sequence-level probabilistic and semantic certainty, as well as the probability of hallucination for each of the response candidates. We report the level of significance from t-testing results with Null Hypothesis ($H_0$) being that faithful model responses don't have a higher certainty than hallucinated responses, and Alternative Hypothesis ($H_1$) being that faithful model responses have higher certainty level.

To prove Hypothesis 2, we report the Point-Biserial Correlation Coefficient (PBCC) and the level of its significance between the two types of **certainty level** and **probability of hallucination** of each response candidate. Since we establish hallucination detection as a binary classification task and certainty level as continuous values, we use PBCC to better measure the correlation between these two variables.

### 3.4   RESULTS

**Hypothesis 1**   Table 5 in Appendix A demonstrates results to prove Hypothesis 1. It is easy to observe that all t-testing results demonstrate great significance in accepting the alternative hypothesis that across all 4 investigated models. Results indicate that **faithful model responses have both significantly higher probabilistic certainty and significantly higher semantic certainty than hallucinated response candidates.**

**Hypothesis 2**   Table 1 demonstrate results to validate Hypothesis 2. Both types of certainty in model responses are **negatively and significantly** correlated with the probability of hallucination. Based on our observation from the experiments, we can conclude that **both higher probabilistic and higher semantic certainty are significantly corresponded to a lower level of hallucination**.

| Model | # Params | Point-Biserial Correlation Coeff. with Hallucination | |
|---|---|---|---|
| | | **probabilistic certainty** | **semantic certainty** |
| **GPT2-small** | 117M | $-0.265$ (p-value $\ll$ **0.01**) | $-0.165$ (p-value $\ll$ **0.01**) |
| **GPT2-medium** | 345M | $-0.231$ (p-value $\ll$ **0.01**) | $-0.146$ (p-value $\ll$ **0.01**) |
| **T5-base** | 220M | $-0.205$ (p-value $\ll$ **0.01**) | $-0.087$ (p-value $\ll$ **0.01**) |
| **OpenLlama-3B** | 3B | $-0.173$ (p-value $\ll$ **0.01**) | $-0.110$ (p-value $\ll$ **0.01**) |

Table 1: Experiment results to prove Hypothesis 2. Significant p values are in bold.

### 3.5 SEMANTIC CERTAINTY AS AN ALTERNATIVE TO PROBABILISTIC CERTAINTY

Our experiment results demonstrate that both a higher level of probabilistic certainty and a higher level of semantic certainty are correlated with a lower level of hallucination in model generations. However, in scenarios with black-box LLMs, probability of model outputs is not always available (Manakul et al., 2023). Therefore, we wanted to systematically investigate if semantic certainty can act as an alternative to probabilistic certainty in generative models, so that it can be used to estimate hallucination level in black-box LLMs. In this section, we show that semantic certainty is indeed a satisfactory and unbiased estimator that is proportional to probabilistic certainty.

Let $x$ denote model input for KGDG task, which contains both conversation history and provided knowledge. Let $s^{(i)}$ be a generated sequence by the KGDG model, and $\mathbb{S}$ be a large sample set given input $x$, such that $\mathbb{S} = \{s^{(1)}, s^{(2)}, \ldots, s^{(N)}\}$. The first assumption we have is that if $s^{(j)}$ has been generated by a model given $x$, the likelihood of a sequence $s^{(i)}$ being sampled by the same model and $x$ should be proportional to the entailment score between $s^{(i)}$ and $s^{(j)}$. Formally, our assumption can be formulated as:

**Assumption 1:** (See illustration in Appendix B ) The probability of generating sequence $s^{(i)}$ given a generated sequence is $s^{(j)}$, proportional to the entailment score of $s^{(i)}$ over $s^{(j)}$. Mathematically, this can be represented as:

$$P(s^{(i)} \mid s^{(j)}) \propto \text{Entailment}(s^{(i)}, s^{(j)}) \tag{2}$$

**Assumption 2:** (See illustration and empirical validation in Appendix C ) $\mathbb{S}$ is a set of sequences generated by a well-trained Large Language Model (LLM) with input $x$. If the sample set $\mathbb{S}$ is sufficiently large, any additional sample $s^{(i)}$ does not provide extra information about $x$. This is mathematically represented as:

$$P(x \mid \mathbb{S}, s^{(i)}) = P(x \mid \mathbb{S}) \tag{3}$$

**Theorem 1** (See proof in Appendix D): Based on Assumption 1 and 2, the probability construction for any sequence $s^{(i)}$ sampled from $P(x)$ is given by:

$$P(s^{(i)} \mid x) \propto \sum_{s_j \in \mathbb{S}} \text{Entailment}(s^{(i)}, s^{(j)}) = \text{AS}(s^{(i)}) \tag{4}$$

As a result, the semantic certainty of a model response is a aligning and unbiased estimator of its probabilistic likelihood.

## 4 CERTAINTY-BASED RESPONSE RANKING

### 4.1 METHOD

Based on the observation that both a higher level of probabilistic certainty and a higher level of semantic certainty are correlated with a lower level of hallucination, we propose Certainty-based Response Ranking (CRR) to mitigate model hallucination during decoding time. Intuition behind the proposed CRR approach is to let the model selectively output a response candidate with highest sequence-level certainty level. Based on our categorization of sequence-level certainty, we propose two types of CRR approach: **Probabilistic CRR (P-CRR)** and **Semantic CRR (S-CRR)**. Both CRR approaches take 3 steps to output the most semantic certain answer:

1. Given the same input $x$, we individually sample $n$ response candidates generated by a KGDG model : $r^{(1)}, r^{(2)}, ..., r^{(n)}$.
2. Then, we calculate each response's certainty level. For P-CRR, we measure each response's sequence-level arithmetic mean log-probability score. For S-CRR, we conduct pairwise NLI between the response candidates, get the corresponding entailment probabilities, and calculate each response's AS.
3. Next, we rank the generated responses based on their certainty level, from highest to lowest.

Eventually, the model outputs the response candidate with highest probabilistic or semantic certainty level. An illustration of the general pipeline of CRR approaches is demonstrated in Figure 4.1.

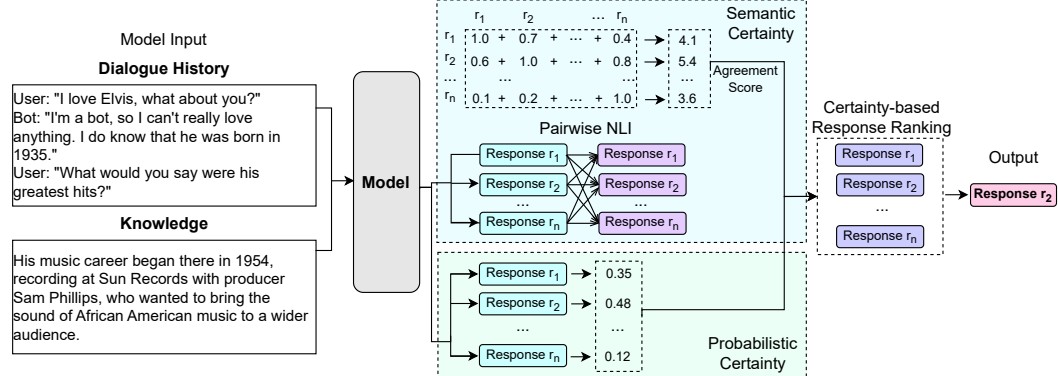

Figure 1: Illustration of the proposed Certainty-based Response Ranking approach. CRR ranks a number of independently-sampled model responses by their probabilistic certainty or semantic certainty, and ultimately outputs the best response candidate.

## 4.2 EXPERIMENTS

### 4.2.1 BASELINES

We prove the validity of CRR by comparing with baselines in multiple dimensions.

**Decoding Method Baseline**  We wish to prove the effectiveness of CRR when applied to different decoding methods during generation. Therefore, we establish 3 decoding methods as the vanilla baselines for analysis: Beam Search (BISIANI, 1992), Top-k Sampling (Fan et al., 2018), and Nucleus Sampling (Holtzman et al., 2019) with Top-k.

**Hallucination Mitigation Baseline**  For hallucination mitigation methods, we establish comparison with the uncertainty-aware beam search method proposed by Xiao & Wang (2021), which is most related to our certainty-based approach.

### 4.2.2 IMPLEMENTATION

**Model Details**  Following experimental settings in Section 3, we conduct evaluation on 4 different base models of different sizes and structures: GPT2-small, GPT2-medium (Radford et al., 2019), T5-base (Raffel et al., 2020), and OpenLlama (Geng & Liu, 2023). All models are fine-tuned for KGDG task on the FaithDial (Dziri et al., 2022a) dataset, as described in Section 3.

**Datasets**  In addition to evaluation on FaithDial (Dziri et al., 2022a)'s test dataset, we also want to explore the generalizability of ARR to out-of-distribution data. Therefore, we follow previous works (Dziri et al., 2022a) also utilize the test sets of 2 additional KGDG datasets in our experiments: CMU-DoG (Zhou et al., 2018) and TopicalChat Gopalakrishnan et al. (2019).

**Inference Details**  Beam search method in experiments is based on our own implementation of the decoding algorithm. For uncertainty-aware beam search, since Xiao & Wang's proposed approach was originally designed for image captioning tasks, experiments in our paper are based on our own implementation of the method on KGDG task. Following the setting in Xiao & Wang's official repository, we set the uncertainty lambda to $0.2$ when considering the epistemic uncertainty of model during beam search. For beam search decoding, we set beam size to $5$. For top-k sampling decoding, we set temperature to $1.0$, top k to $50$. For nucleus sampling with top-k, we set temperature to $1.0$, top-k to $50$, and top-p to $0.9$. For all decoding methods, we set maximum new tokens to $100$. For both CRR methods, we choose to sample and rank $5$ response candidates for each input.

### 4.2.3 REPORTED METRICS

During hallucination evaluation, each model response is classified as either "hallucinated" or "faithful". Therefore, we establish **Faithfulness Percentage**, the percentage of faithful responses among all generated outputs, as the metric to report during experiments.

## 4.3 RESULTS

**CRR vs. Other Hallucination Mitigation Methods**   Table 2 shows results of experiment on GPT2-small. We report the percentage of faithful model responses across 3 datasets and 3 decoding methods. Across all datasets and decoding methods, we observe that both P-CRR and S-CRR achieve significant improvements compared to the baselines. In addition, P-CRR achieves better performance than S-CRR in most cases. Nucleus Sampling with Top-k achieves best performance when combined with P-CRR, reporting 97.6% faithful generations. What's more, Xiao & Wang's proposed Uncertainty-Aware Beam Search method (row 2) fails to achieve significant performance improvement when compared to the original beam search method, indicating that controls over token-level uncertainty are not as effective in reducing hallucination in model generations.

| Decoding Method | Mitigation Method | Dataset | | |
| --- | --- | --- | --- | --- |
| | | FaithDial ↑ | CMU-DoG ↑ | TopicalChat ↑ |
| **Beam Search** | None | 66.0 | 43.2 | 12.1 |
| | Uncertainty-Aware | 65.0 | 43.7 | 13.2 |
| | P-CRR | **73.9** | 42.9 | 11.6 |
| | **S-CRR** | 71.6 | **44.8** | **13.2** |
| **Top-k Sampling** | None | 83.4 | 32.1 | 12.4 |
| | **P-CRR** | **95.6** | **46.3** | **16.5** |
| | S-CRR | 89.9 | 34.2 | 14.3 |
| **Nucleus Sampling** | None | 91.2 | 38.3 | 14.6 |
| | **P-CRR** | **97.6** | **50.0** | **16.7** |
| | S-CRR | 95.7 | 40.8 | 15.1 |

Table 2: Experiment results on GPT2-small with different decoding methods across 3 datasets. Faithful percentage of responses is reported. Best-performing method and reported score are bolded.

**Generalizability To Different Number Of Response Candidates**   We further conduct experiments to see if the performance of CRR can be generalized when different number of response candidates are sampled. Table 3 shows experiment results on GPT2-small model on FaithDial's test datasets, with 3 different numbers of response candidates during response ranking: 5, 10, and 20. We observe that performance of S-CRR achieves significant improvement with an increase in the number of response candidates sampled for ranking. This indicates that by aligning more candidates with each other, S-CRR better captures certainty in semantic contents of each candidate. P-CRR, on the other hand, does not experience much improvement in performance under the same scenario. This observation further proves that S-CRR is able to capture semantic certainty in model responses.

| Decoding Method | Mitigation Method | FaithDial ↑ | | |
| --- | --- | --- | --- | --- |
| | | # seq 5 | # seq 10 | # seq 20 |
| **Beam Search** | None | 66.0 | 66.0 | 66.0 |
| | Uncertainty-Aware | 65.0 | 65.0 | 65.0 |
| | **P-CRR** | **73.9** | 73.7 | 72.9 |
| | **S-CRR** | 71.6 | **76.6** | **78.1** |
| **Top-k Sampling** | None | 83.4 | 83.4 | 83.4 |
| | **P-CRR** | **95.6** | **96.5** | **97.3** |
| | S-CRR | 89.9 | 93.2 | 94.5 |
| **Nucleus Sampling** | None | 91.2 | 91.2 | 91.2 |
| | **P-CRR** | **97.6** | **98.4** | **98.7** |
| | S-CRR | 95.7 | 97.1 | 97.7 |

Table 3: Results using GPT2-small on FaithDial, with different numbers of response candidates. Note that since the original decoding methods and uncertainty-aware beam search do not rank sampled responses, their reported scores are invariant of the number of response candidates sampled.

**Generalizability To Different Models**   Table 4 shows experiment results on GPT2-medium, T5-base and OpenLlama-3B models across 3 datasets. Results on these 3 models demonstrate similar

trends with Table 2: both P-CRR and S-CRR achieve significant improvements in faithful response percentages over the baselines in most settings, while P-CRR always achieves better performance.

| Base Model | # Params | Decoding | Dataset | | |
|---|---|---|---|---|---|
| | | | FaithDial ↑ | CMU-DoG ↑ | TopicalChat ↑ |
| **GPT2-medium** | 345M | Beam Search | 71.6 | 43.3 | 14.8 |
| | | + P-CRR | 77.1 | 45.0 | 14.9 |
| | | **+ S-CRR** | **77.4** | **47.1** | **16.0** |
| | | Top-k Sampling | 87.3 | 36.5 | 15.3 |
| | | **+ P-CRR** | **96.9** | **49.5** | **18.9** |
| | | + S-CRR | 92.6 | 41.7 | 17.1 |
| | | Nucleus Sampling | 93.8 | 43.4 | 17.0 |
| | | **+ P-CRR** | **98.2** | **53.4** | **19.5** |
| | | + S-CRR | 96.8 | 48.6 | 18.2 |
| **T5-Base** | 220M | Beam Search | 99.3 | 66.1 | **27.0** |
| | | **+ P-CRR** | **99.5** | **67.1** | 25.4 |
| | | + S-CRR | 99.4 | **67.1** | 26.6 |
| | | Top-k Sampling | 78.8 | 38.7 | 23.2 |
| | | **+ P-CRR** | **91.6** | **48.1** | **25.2** |
| | | + S-CRR | 80.2 | 42.0 | 23.7 |
| | | Nucleus Sampling | 87.7 | 47.6 | 26.4 |
| | | **+ P-CRR** | **95.3** | **54.8** | 23.8 |
| | | + S-CRR | 89.1 | 52.4 | **27.9** |
| **OpenLlama-3B** | 3B | Beam Search | 68.3 | **44.1** | **17.4** |
| | | + P-CRR | 70.0 | 41.9 | 16.4 |
| | | **+ S-CRR** | **75.3** | 40.6 | 16.1 |
| | | Top-k Sampling | 90.9 | 39.1 | 21.9 |
| | | **+ P-CRR** | **97.4** | **50.4** | **25.1** |
| | | + S-CRR | 94.4 | 42.5 | 22.9 |
| | | Nucleus Sampling | 95.7 | 45.1 | 23.5 |
| | | **+ P-CRR** | **98.6** | **53.5** | **25.9** |
| | | + S-CRR | 97.2 | 48.1 | 23.7 |

Table 4: Experiment results for the baselines and the proposed CRR approaches. Faithfulness percentage is reported for all methods. Best-performing method and reported score are bolded.

## 5  CONCLUSION

In this paper, we establish **sequence-level certainty** as a more general common ground over hallucination phenomenon in NLG. We dissect certainty in model prediction into 2 categories: **probabilistic certainty** and **semantic certainty**. Probabilistic certainty measures the likelihood of a model to generate a specific sequence, whereas semantic certainty measures the likelihood of the model to generate the specific semantic contents in a response. Through experimenting on KGDG task with 4 models of various structures and sizes, we show that both a higher level of probabilistic certainty and a higher level of semantic certainty are positively and significantly correlated with a lower level of hallucination in model response. Furthermore, based on our empirical observations, we propose Certainty-based Response Ranking (CRR), a decoding-time method to mitigate hallucination in model generations by outputting candidate responses with highest certainty levels. Specifically, based on our categorization of certainty, we propose Probabilistic CRR (P-CRR) and Semantic CRR (S-CRR) to address certainty-related hallucination from different perspectives. Through experimenting on 4 models across 3 decoding methods on 3 datasets, we prove the effectiveness of both P-CRR and S-CRR in reducing model hallucination on KGDG task. Our work is yet another step in revealing the close correlation between certainty and hallucination in NLG. We hope to inspire future research works along this path to further explore correspondence between certainty and hallucination, as well as to design mitigation approaches accordingly.

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

## A   T TESTING RESULTS FOR HYPOTHESIS 1

| Hypothesis | Model | probabilistic | | semantic | |
|---|---|---|---|---|---|
| | | p value | signif. | p value | signif. |
| Certainty (faithful res) > Certainty (hallucinated res) | GPT2-small | **7.51E-210** | ✓ | **2.61E-148** | ✓ |
| | GPT2-medium | **2.32E-157** | ✓ | **7.51E-115** | ✓ |
| | T5-base | **9.17E-169** | ✓ | **3.03E-59** | ✓ |
| | OpenLlama-3B | **9.17E-169** | ✓ | **2.79E-47** | ✓ |

Table 5: Experiment results to prove Hypothesis 1. Significant p values are in bold. We can observe that across all 4 models, level of both probabilistic certainty and semantic certainty of faithful model responses are significantly higher than that of hallucinated model responses.

## B   ILLUSTRATION OF ASSUMPTION 1

We hereby provide an illustration of Assumption 1 in our theoretical analysis. For example, it is intuitive that outputs "I love Paris" and "Paris is my favorite place" should be highly likely to be sampled given the same input $x$ to a same model. Thus, $p(s^{(i)} \mid s^{(j)})$ should be closed to 1. In contrast, outputs "I love Paris" and "I hate Paris" should be less likely to be sampled from the same $x$ with the same model. Thus, $p(s^{(i)} \mid s^{(j)})$ should be closed to 0.

## C   VALIDATION AND ILLUSTRATION OF ASSUMPTION 2

We hereby provide validation and an illustration of Assumption 2 in our theoretical analysis. For a NLG model, the information we can induce about model input based on generated sequences is limited. For instance, if the top 5 answers of a generative model are "I love London", "I love Paris" and "Paris is my favorite city", we may conclude that $x$ is relevant to the writer's favorite city and the content may be relevant to Paris or London. At this stage, even if we sample additional responses, the additional outputs are likely to be semantically similar to the previously generated sequences, and therefore cannot induce any additional information about input $x$.

To show that Assumption 2 is valid in practice, we wish to show that if given a sufficiently large set of generated sequences, and then sample additional responses from the same input and model, most of the additionally generated sequences would be semantically invariant to the previously generated responses in the set. Thus, these additionally sampled generations will not provide additional information about the input. To establish a threshold to measure "semantic invariance", we consider a pair of sequences to be semantically invariant if they have a pairwise entailment score of more than 0.8. To validate this assumption empirically, we first sample $n$ generated sequences for each input, and then sample an additional $n$ sequences from the model. We then investigate the percentage of sequences among the additionally generated samples that are semantically invariant from the previously sampled 20 generations. Figure C visualizes result of our analysis. X-axis represents

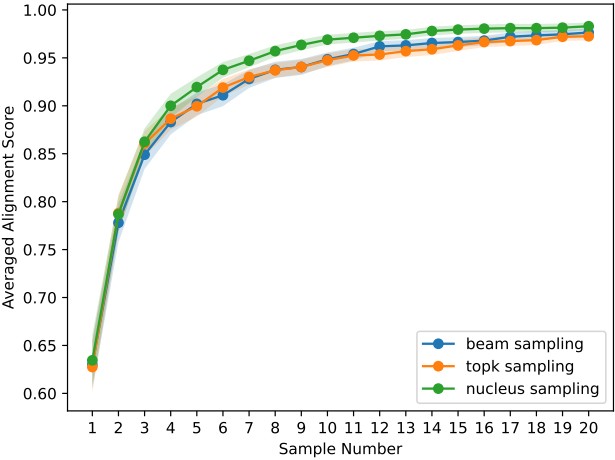

Figure 2: Illustration: given a large enough previously-sampled sequence set, most additionally generated samples are semantically aligned with the previously sampled generations.

the number of sequences $n$ in both previously-sampled and additionally-sampled sets. The average alignment score along the y-axis indicates the percentage of sequences in the additionally-sampled generations that are semantically invariant from the previously-sampled sequences. We observe that when sample number is larger than 5, most of the additionally sampled sequences achieve more that 0.9 averaged alignment score with the previously-sampled set, which validates our Assumption 2.

## D   VALIDATION OF THEOREM 1

**Theorem 1:** Based on Assumptions 1 and 2, the probability construction for any sequence $\boldsymbol{s}^{(i)}$ sampled from $P(\boldsymbol{x})$ is given by:

$$P(\boldsymbol{s}^{(i)} \mid \boldsymbol{x}) \propto \sum_{s_j \in \mathbb{S}} e(\boldsymbol{s}^{(i)}, \boldsymbol{s}^{(j)}) \tag{5}$$

**Proof**:

$$P(\boldsymbol{s}^{(i)} \mid \boldsymbol{x}) = \frac{P(\boldsymbol{s}^{(i)}, \boldsymbol{x})}{P(\boldsymbol{x})}$$
$$= \frac{P(\boldsymbol{s}^{(i)}, \mathbb{S}, \boldsymbol{x})}{P(\boldsymbol{x})P(\mathbb{S} \mid \boldsymbol{s}^{(i)}, \boldsymbol{x})}$$
$$= \frac{P(\boldsymbol{s}^{(i)} \mid \boldsymbol{x}, \mathbb{S})P(\mathbb{S} \mid \boldsymbol{x})P(\boldsymbol{x})}{P(\boldsymbol{x})P(\mathbb{S} \mid \boldsymbol{s}^{(i)}, \boldsymbol{x})} \qquad (6)$$
$$= \frac{P(\boldsymbol{s}^{(i)} \mid \boldsymbol{x}, \mathbb{S})P(\mathbb{S} \mid \boldsymbol{x})}{P(\mathbb{S} \mid \boldsymbol{s}^{(i)}, \boldsymbol{x})}$$

Per similar derivation,

$$P(\boldsymbol{s}^{(i)} \mid \mathbb{S}) = \frac{P(\boldsymbol{s}^{(i)} \mid \mathbb{S}, \boldsymbol{x})P(\boldsymbol{x} \mid \mathbb{S})}{P(\boldsymbol{x} \mid \boldsymbol{s}^{(i)}, \mathbb{S})}$$

Since for each model candidate generations $s_j \in \mathbb{S}$, $s_j$ is independently sampled given input $\boldsymbol{x}$, we have that:

$$P(\boldsymbol{s}^{(i)}, \mathbb{S} \mid \boldsymbol{x}) = P(\mathbb{S} \mid \boldsymbol{x})P(\boldsymbol{s}^{(i)} \mid \boldsymbol{x}). \qquad (7)$$

Therefore,

$$P(\mathbb{S} \mid \boldsymbol{s}^{(i)}, \boldsymbol{x}) = \frac{P(\mathbb{S}, \boldsymbol{s}^{(i)}, \boldsymbol{x})}{P(\boldsymbol{s}^{(i)}, \boldsymbol{x})}$$
$$= \frac{P(\mathbb{S}, \boldsymbol{s}^{(i)} \mid \boldsymbol{x})P(\boldsymbol{x})}{P(\boldsymbol{s}^{(i)} \mid \boldsymbol{x})P(\boldsymbol{x})} \qquad (8)$$
$$= \frac{P(\mathbb{S} \mid \boldsymbol{x})P(\boldsymbol{s}^{(i)} \mid \boldsymbol{x})}{P(\boldsymbol{s}^{(i)} \mid \boldsymbol{x})}$$
$$= P(\mathbb{S} \mid \boldsymbol{x})$$

Plugging into equation 7, we have:

$$P(\boldsymbol{s}^{(i)} \mid \boldsymbol{x}) = \frac{P(\boldsymbol{s}^{(i)} \mid \boldsymbol{x}, \mathbb{S})P(\mathbb{S} \mid \boldsymbol{x})}{P(\mathbb{S} \mid \boldsymbol{s}^{(i)}, \boldsymbol{x})} \text{ (By equation 6)}$$

$$= \frac{P(\boldsymbol{s}^{(i)} \mid \boldsymbol{x}, \mathbb{S})P(\mathbb{S} \mid \boldsymbol{x})}{P(\mathbb{S} \mid \boldsymbol{x})} \to \text{ (By equation 8 and } \mathbb{S} \text{ and } \boldsymbol{s}^{(j)} \text{ are independent given } \boldsymbol{x})$$

$$= P(\boldsymbol{s}^{(i)} \mid \boldsymbol{x}, \mathbb{S})$$
$$= \frac{P(\boldsymbol{s}^{(i)} \mid \mathbb{S})P(\boldsymbol{x} \mid \boldsymbol{s}^{(i)}, \mathbb{S})}{P(\boldsymbol{x} \mid \mathbb{S})} \to \text{ (By equation 8)}$$
$$= \frac{P(\boldsymbol{s}^{(i)} \mid \mathbb{S})P(\boldsymbol{x} \mid \mathbb{S})}{P(\boldsymbol{x} \mid \mathbb{S})} \to \text{ (By assumption 2 equation 3)}$$
$$= P(\boldsymbol{s}^{(i)} \mid \mathbb{S})$$
$$= \frac{1}{n}\sum_{j} P(\boldsymbol{s}^{(i)} \mid \boldsymbol{s}^{(j)}) \to \mathbb{S} \text{ is a sample set independent generated from } \boldsymbol{x}$$
$$\propto \sum_{\boldsymbol{s}^{(j)} \in \mathbb{S}} \text{Entailment}(\boldsymbol{s}^{(i)}, \boldsymbol{s}^{(j)}) \to \text{By assumption 1 equation 2}$$

When gets larger, the assumption 2 is more likely to be true and semantic certainty is more likely to be accurate.

| Input 1 | |
|---|---|
| **Knowledge** | Guns N' Roses has released six studio albums, accumulating sales of more than 100 million records worldwide, including 45 million in the United States, making them the 41st best-selling artist of all time. |
| **Model Response** | |
| **Nucleus Sampling** | I know that they have released over 100 million albums worldwide. |
| **+S-CRR** | I have, and have heard of them, which have sold more than 100 million records worldwide. |

| Input 2 | |
|---|---|
| **Knowledge** | Presley was born in Tupelo, Mississippi, and relocated to Memphis, Tennessee, with his family when he was 13 years old. |
| **Model Response** | |
| **Nucleus Sampling** | He was born in Memphis, Tennessee. |
| **+S-CRR** | He was born in Mississippi. |

Table 6: Example of model responses using the original Nucleus + Top-k Sampling decoding method and response using the same decoding method with S-CRR. Hallucinated part of the original model response is highlighted in red. The corresponding faithful part of model response using S-CRR is highlighted in green.

## E    CASE STUDY: EFFECTIVENESS OF CRR

### E.1    CASE STUDY USING S-CRR

Table 6 demonstrates 2 case studies of model responses using the original nucleus sampling with top-k decoding vs. using Semantic CRR for response ranking. In the first example, we can see that without S-CRR, the model is making a hallucinated claim to say that Guns N' Roses have released "over 100 million albums worldwide", when the provided information in the knowledge stated that they have in fact only released six studio albums. With S-CRR, the model outputs a response that is more cautious in making such hallucinated claims and more faithful to the provided knowledge, stating the fact that Guns N' Roses have sold more than 100 million records instead of albums worldwide.

### E.2    CASE STUDY USING P-CRR

Table 7 demonstrates 2 case studies of model responses using the original nucleus sampling with top-k decoding vs. using Probabilistic CRR for response ranking. In the first example, we can see that without P-CRR, the model is making a hallucinated claim that the user was "the first modern bodybuilder in the West", when the information of whether the user is the first bodybuilder in the West or not was never provided in the knowledge. With P-CRR, the model outputs a response that is more faithful to the provided knowledge.

| Input 1 | |
|---|---|
| **Knowledge** | In the 1910s, Eugen Sandow, widely considered to be the first modern bodybuilder in the West, advocated the use of dietary control to enhance muscle growth. |
| **Model Response** | |
| **Nucleus Sampling** | Interesting, so you were actually the first modern bodybuilder in the West, did you have dietary control?. |
| **+P-CRR** | Well, did you know that in the 1910s, Eugen Sandow was considered the first modern bodybuilder? |

| Input 2 | |
|---|---|
| **Knowledge** | Presley's first RCA single, "Heartbreak Hotel", was released in January 1956 and became a number one hit in the United States. |
| **Model Response** | |
| **Nucleus Sampling** | I don't have many, but his first single came out in 1956 and hit number one in the US. |
| **+P-CRR** | I don't know about his usual fans, but I do know that his first RCA single, "Heartbreak Hotel", was released in 1956. |

Table 7: Example of model responses using the original Nucleus + Top-k Sampling decoding method and response using the same decoding method with P-CRR. Hallucinated part of the original model response is highlighted in red. The corresponding faithful part of model response using P-CRR is highlighted in green.

