# OpenReview forum: "Sequence-Level Certainty Reduces Hallucination In Knowledge-Grounded Dialogue Generation"
_ICLR.cc/2024/Conference — ICLR 2024 Conference Withdrawn Submission_

### Official Review · Reviewer_Ck8E · 2023-10-24

**Soundness:** 2 fair
**Presentation:** 2 fair
**Contribution:** 2 fair
**Rating:** 5
**Confidence:** 4

**Summary:**

The paper explores how to reduce hallucination in text generation models. It makes the claim that previous methods have focused on token level checks, and claims novelty by instead focussing on sentence (sequence) level scores. In particular it introduces 2 measures:
1: average of token level log-probabilities
2. agreement of an entailment model of candidate decoding compared to several other options decoded via some decoding method.

These two measures are then used to guide decoding, or selection of decoded utterances, in order to reduce hallucination.
Results are given claiming that these measures a helpful at detecting hallucination, and helpful at preventing hallucination when used in decoding strategy.

**Strengths:**

An important problem, with a meaningful contribution given in this paper, particularly in using entailment across candidates (ie consistency) to detect hallucinations.

**Weaknesses:**

* The reported metrics section (4.2.3) introduces a faithfulness percentage. This is a clear and obvious measure ... however it isn't described how this is calculated! I presume it was manually done, rather than inferred by some model or automatic metric? If so who manually graded this?

* It is very odd that top-k and top-p sampling methods result in reducing hallucination in a knowledge-grounded dialog generation. There is a lack of detail on what the actual task is here, and although the FaithDial dataset is referenced, it would help the reader a lot to actually describe what precise task is being addressed. There are comments as well about hallucinating in wider tasks like dialog generation, which I take to mean NLG tasks not grounded on factual information in the inputs. If this is so, how is hallucination even defined there?

* Results for the number 1 hypothesis should not be in the appendix!
* I'm unclear if the statistical tools used to make claims during
* What is the entailment model used in measuring equation 1? How important is this to the final results? Does changing the entailment model result in entirely different outcomes, presumably so.
* The writing is poor in that it repeats sections multiple times. For example probabilistic and semantic certainty are introduced 3 or 4 times.

**Questions:**

* Interested for the authors thoughts on why likelihoods are based on product of of token probs, but uncertainty is better detected via mean of token (log) probs?
* Were other stats beyond the mean (min, max, variance, etc) considered in forming p-crr?

---

> ### Author Response · Authors · 2023-11-21
> **Thanks & Response to Reviewer (1/2)**
>
> We thank the reviewer for recognizing the novelty of the proposed approach and our contribution to research on model hallucinations. Below, we address your questions.
> ### **[Q1 - How is Faithfulness Percentage calculated? ---- We use FaithCritic, a State-Of-The-Art hallucination classification model for KGDG task, for identifying hallucinated responses. We then report the percentage of the hallucinated responses on the test set.]**
>
> In our experiments, we follow previous work [1] to report the faithfulness percentage of model responses using FaithCritic, a hallucination classification model. In our paper, we discussed our choice of hallucination classification model on page 4, Section 3.3.1. In section 4.2.2, we stated that we followed model selection experiment settings from Section 3. For better readability, we will re-emphasize this point in Section 4.2.2 in the final version.
>
> ### **[Q2 - How is 'Hallucination' defined here? ---- We follow previous works and define hallucination on KGDG task to be responses with information that is unattributable to the provided knowledge in model input. ]**
>
> We defined the KGDG task and the definition of hallucination in KGDG **in the first paragraph of our introduction section**. We hereby provide a clarification on the task formulation and hallucination definition.
> * In KGDG, a dialogue model is provided with textual knowledge and a series of conversation histories, and is expected to generate informative and meaningful responses to the previous conversation with the provided knowledge.
> * Previous work [2] on hallucination problems in KGDG task provides the following definitions of faithfulness and hallucination on this task:
>   * Factually Consistent Response is consistent with the provided knowledge.
>   * Hallucinated Response is not consistent with the knowledge but may still be correct.
> * We follow related previous works [1] [2] [3] to define hallucinations in this task as containing information that is unattributable to the source knowledge provided to the model.
> * In order to better convey this information to readers, we will add a ‘task definition’ subsection in the experiment section in the final version of our paper to highlight the details.
>
> ### **[Q3 - Results for the number 1 hypothesis should not be in the appendix! ---- Thank you for recognizing the validity of our experiment result! We will make spaces to include it in the main paper for the final version.]**
>
> Regarding your point on the results for hypothesis 1, we were only able to include the table in the appendix due to the page limit. For the final version, we will make necessary modifications to include it in the main paper in the final version.
>
> ### **[Q4 - Statistical Tool Usage? ---- There’s potentially a compiling/formatting error. We are happy to further address your question if you can provide more details.]**
>
> We are not able to view your full question in this line, potentially due to a compiling or formatting error. Would appreciate it if the reviewer could provide more details on the question!

---

> ### Author Response · Authors · 2023-11-21
> **Response to Reviewer (2/2)**
>
> ### **[Q5 - What is the entailment model used and how important is this? ---- We use an off-the-shelf Roberta Large-based Natural Language Inference (NLI) model. An equally strong NLI model should yield similar results.]**
>
> * We used **an off-the-shelf Roberta Large-based Natural Language Inference (NLI) model** as the entailment model. We then calculate the **Entailment Score** as **the probability of being classified as 'entailed'** by the NLI model.
>   * We discussed our choice of the entailment model **on page 4, Section 3.3**, and said in Section 4 that we 'followed model selection experiment settings from Section 3'. In the final version, we will include full details again in Section 4.2.2 to re-emphasize this information to users.
> * Regarding your question on the importance of the entailment model to the final results, we definitely agree that **choosing an NLI model with high performance is crucial to delivering satisfactory results**, since a good entailment model **will better recognize and quantify semantic agreement** between response candidates.
>   * Theoretically, changing to another entailment model with **equally high performance** should **not** result in drastically different outcomes. On the other hand, changing to a **poor-performing** entailment model **might** result in poor results, since the entailment model would not be able to correctly identify and calculate semantic agreement and therefore semantic certainty of response candidates.
>
> ### **[Q6 - Repeats in Writing. ---- Thanks for pointing it out! We will modify our writing to be more concise in the final version.]**
>
> ### **[Q7 - Uncertainty as the mean of token log probabilities. ---- We believe that both are probability-based approaches to measure certainty in model responses, we provide details of the log probability-based certainty measure below.]**
>
> Regarding your question on likelihood and log probabilities, we believe that both are probability-based approaches that can measure certainty in model responses. We formulate **sequence-level probabilistic certainty as the mean of token log probability** because this **better measures the probability of generating the whole sequence**. For instance, given a context $c$ and a sequence $y_1 y_2 y_3$, the probability of generating the sequence would be $(p(y_1|c)p(y_2|y_1, c)p(y_3|y_1,y_2,c))$, which, when taken $log$, would be:
> $$
> log(p(y_1|c)p(y_2|y_1, c)p(y_3|y_1,y_2,c)) = log p(y_1|c) + log p(y_2|y_1, c) + log p(y_3|y_1,y_2,c).
> $$
>
> ### **[Q8 - Other stats for P-CRR? ---- We did not explore the usage of other stats except the mean for probabilistic-based uncertainty measure, but it could be worth exploring in future works.]**
>
> Regarding your question on the usage of other stats in forming P-CRR, **no other stats beyond the mean** were considered since **we wanted to measure the overall sequence-level certainty** of a generated response. However, it is definitely an interesting direction to explore in future works. For instance, would a high variance in sequence-level probabilities be associated with hallucination levels? We belive that the contributions of our study lays a solid foundation for future explorations along this direction.

---

> > ### Author Response · Authors · 2023-11-21
> > **Referenced Work**
> >
> > [1] Nouha Dziri et al. FAITHDIAL: A Faithful Benchmark for Information-Seeking Dialogue. TACL 2022.
> >
> > [2] Sashank Santhanam et al. Rome was built in 1776: A Case Study on Factual Correctness in Knowledge-Grounded Response Generation. CoRR 2021.
> >
> > [3] Hannah Rashkin et al. Increasing Faithfulness in Knowledge-Grounded Dialogue with Controllable Features. ACL 2021.

---

### Official Review · Reviewer_qG74 · 2023-10-27

**Soundness:** 2 fair
**Presentation:** 2 fair
**Contribution:** 2 fair
**Rating:** 5
**Confidence:** 4

**Summary:**

The authors propose a sequence-level certainty to addresses model hallucinations in Natural Language Generation (NLG). Expecially they demonstrate a significant correlation between probabilistic certainty (perplexity of the generated tokens) and semantic certainty (the entitlement score between all the pairs of generated response) in model responses and hallucination metrics. They introduce two Certainty-based Response Ranking (CRR) methods, Probabilistic CRR (P-CRR) and Semantic CRR (S-CRR), which effectively mitigate hallucination in NLG across various datasets and models.

**Strengths:**

Strengths:

- The paper introduces an interesting approach that addresses model hallucination in Natural Language Generation (NLG).
- The experimental methodology is rigorous, providing statistical significance in the context of NLG tasks.
- The paper offers fair comparisons with existing hallucination reduction decoding methods.

**Weaknesses:**

Weaknesses:

- The paper's scope is limited as it focuses on a specific NLG task, knowledge-grounded dialogue responses. In this setting, hallucination are not very pronounced, since the gold document is provided as input to the model. I suspect that a larger LLM (prompted correctly) can improve any of the provided metric  without any "special" decoding method (greedy). I invite the authors to report this baselines if possible.

**Questions:**

Check weaknesses.

**Details Of Ethics Concerns:**

nan

---

> ### Author Response · Authors · 2023-11-21
> **Thanks & Response to Reviewer**
>
> We thank the reviewer for recognizing the novelty and validity of the proposed approach and experiment designs. Below, we address your concern.
> ### **[Q - Prompting a larger LLM might improve metric. ---- We proved the effectiveness of our method on LLMs through experimenting with OpenLlama, a representative of the family of LLMs. Since our task requires full fine-tuning of models during experiments, results by prompting larger black-box models are of a different experiment setting and also might vary with different prompt choices.]**
>
> * Regarding your point on the scope of the paper, we believe that our study **utilizes KGDG task as a lens** to study **the correlation between sequence-level probabilistic and semantic certainty and level of hallucinations** in generative language models.
>   * Although hallucinations on KGDG task might seem pronounced as the reference knowledge is provided in the input, the simplicity of this task is also its strength: it **helps isolate analysis of hallucinations** in this task, which are responses that are unattributable to the provided knowledge. In addition, as we revealed in our experiments, **recently-developed LLMs such as OpenLlama still hallucinates on this simple task**.
>   * Given the page limit, we were not able to extend our study to other tasks, but our proposed methods and concepts **lay great foundations for future work** along this direction.
> * Regarding your point on experiment results using LLMs, we **conducted experiments with the OpenLlama model**, a representative of the recent LLM family. Positive experiment results **prove the effectiveness of our method when scaled to bigger generative LLMs**.
> * However, we were not able extend to models with even larger models due to the **constraints on training resources**, since we fine-tune all models on the KGDG task.
> * We also did not extend experiments to black-box models such as ChatGPT because our experiments require full fine-tuning of all models on KGDG task, which **cannot be achieved on these API-only models**. We would need to use prompting to get model answers on this task, and there would be **no guarantee of the model’s instruction-following ability for every response**. Model **performances might also vary based on the prompt choices** we use for the task.
>   * In addition, **this experiment setting would diverge drastically** from the other models, since all reported models were fine-tuned to conduct the KGDG task without prompting.

---

### Official Review · Reviewer_drdo · 2023-10-29

**Soundness:** 3 good
**Presentation:** 1 poor
**Contribution:** 2 fair
**Rating:** 3
**Confidence:** 3

**Summary:**

This work tries to reduce the hallucination in NLGs.  To this end, this work has done the following jobs:

1. Based on the previous token-level uncertainty work, this work has defined sentence-level probabilistic certainty and semantic certainty as indicators to evaluate the level of hallucination in model generations. Then, experiments verified the effectiveness of the sentence-level probabilistic certainty and semantic certainty.

2. This work provides theoretical proof to show that the black-box semantic certainty is a good estimator of the white-box probabilistic certainty.

3. To mitigate the hallucination, this work proposes a Certainty-based Response Ranking (CRR), which re-ranks the response according to the sentence-level probabilistic certainty or the sentence-level semantic certainty.

**Strengths:**

1. This work has revealed that both a higher level of probabilistic certainty and a higher level of semantic certainty are significantly correlated with a lower level of hallucination in model generations. It brings an insight for us to reduce the hallucinations in NLGs.

2.  As a white-box metric, probabilistic certainty can not be calculated in practice.  To this end, the authors find that the semantic certainty of a model response is an aligning and unbiased estimator of its probabilistic likelihood.

3. Based on their findings on  Probabilistic certainty and Semantic certainty, this work proposes two ranking methods P-CRR and S-CRR to improve the faithfulness via re-ranking. Experiments can demonstrate their effectiveness, especially the black-box S-CRR.

**Weaknesses:**

1. The proposed methods may significantly damage the diversity of the generated text. In the proposed Certainty-based framework, both P-CRR and S-CRR encourage the backbone model to generate safe, receptive, general responses. Safe responses have higher probability certainty in either a language model or a conditional language model and are always similar to each other. For example, if a model generates three samples, 'I don't know.' , 'I don't know it.' , 'I do not know it.'; then,  both the P-CRR and S-CRR will give higher scores although they are boring.

2. The organization of this work is hard to follow and verbose:

-  As an important concept,  `uncertainty' lacks enough introduction when it first appears. Even in Sec 2.1, the authors only gave a simple literal definition.  Readers can't understand what it is until the 4th page. Meanwhile,  there is no formulation to introduce the token-level uncertainties.

- The authors tended to repeatedly introduce the same thing (for example,  findings proposed by (Xiao & Wang, 2021)) in different sections.

-  In Equation 1, we can not directly infer what $Entailment( *,*)$ is from the nearby context.  The authors only give a very rough Introduction in Sec 3.3.1 without any formulation.

3. The evaluation of dialogues (for example, Table 2) only considers faithfulness. It is necessary to report other metrics (BLEU, ROUGE, DISTINCT, etc.) as well.

**Questions:**

Can you report results on more metrics?

---

> ### Author Response · Authors · 2023-11-21
> **Thanks & Response to Reviewer (1/3)**
>
> We thank the reviewer for recognizing the novelty and effectiveness of the proposed sequence-level semantic certainty and our mitigation method design. Below, we address your questions.
>
> ### **[Q1 - The proposed methods may damage generation diversity. ---- Theoretically, CRR shouldn't hurt diversity since it does not modify models' output distributions. We prove this by experimenting with a GPT2-based model on Text Summarization task. Model performance does not drop with CRR.]**
>
> * Theoretically, the proposed P-CRR and S-CRR methods **will not damage the diversity / quality** of the generated text, because our methods **do not modify the model’s output distribution**. The proposed CRR methods simultaneously and independently sample a number of full responses from a model, and rank them based on their sequence-level probabilistic certainty or semantic certainty. Therefore, there should be **no interference on the model’s generation quality**.
> * In order to further prove our point, we conducted an ablation study on a GPT2-based model for Text Summarization task. The model is inference on the first 1,000 entries of the test split of the **CNN Daily Mail** Text Summarization dataset, and model responses are evaluated using **ROUGE** and **BLEU**, which are text overlap-based metrics. In order to show that CRR does not damage generation quality, we first evaluate the baseline model when **CRR is not applied**. Then, we compare model performance with that **with P-CRR and S-CRR applied**, respectively. The results are as follows:
> | Model              | BLEU | | | ROUGE-1 | | | ROUGE-L | | |
> | :---------------- | :------: | :------: | :------: | ----: |:------: | :------: | :------: | :------: | :------: |
> |   |   **Baseline**   | **+P-CRR** | **+ S-CRR** |    **Baseline**   | **+P-CRR** | **+ S-CRR** |   **Baseline**   | **+P-CRR** | **+ S-CRR** |
> | GPT2-Small           |   **2.80** | 2.70 | 2.79 | 21.60 | 20.89 | **21.72** | 14.25 | 13.46 | **14.56** |
>
> * As observed from the table, applying P-CRR and S-CRR **does not lead to performance drop** on the summarization task. Therefore proving our expectation that CRR **does not damage generation diversity / quality**.

---

> ### Author Response · Authors · 2023-11-21
> **Response to Reviewer (2/3)**
>
> ### **[Q2 - Writing can be more organized and less verbose. ---- Thanks for pointing out! We will make modifications to address these issues in the final version.]**
> * Regarding your points on the presentation of our paper, we will modify the parts of the papers accordingly to better convey our message.
> * Specifically, regarding your question on the definition of $Entailment(\cdot, \cdot)$ in Equation 1, it represents **the probability of being classified as ‘entailed’** by a Natural Language Inference (NLI) model. We will clarify all necessary details in the final version.

---

> ### Author Response · Authors · 2023-11-21
> **Response to Reviewer (3/3)**
>
> ### **[Q3 - Evaluation only reports faithfulness. ---- We report faithfulness using a State-Of-The-Art hallucination classification model, since previous works have pointed out that overlap-based metrics might fail to reflect hallucinations. We re-evaluate model generations on BLEU and ROUGE and obtain positive results.]**
>
> * Regarding your points on the reported metric in this paper, previous studies have pointed out that **traditional overlap-based metrics such as ROUGE and BLEU fail to reflect the hallucination levels** of a model response[2][3][4][5][6][7]. For instance, it has been discovered in abstractive summarization task that:
> > Even if a summary contains a large amount of hallucinatory content, it can achieve a high ROUGE score [2].
> * In KGDG task, we believe these traditional metrics could suffer from two major drawbacks:
>   * They directly compare with a ground truth answer and not the provided knowledge.
>   * They fail to take into account language variations in responses, even when they could mean the same thing as the gold response.
> * Therefore, we chose to report faithfulness scores using FaithCritic, a SOTA NLI-based hallucination classification model, over traditional metrics in our study.
> * However, we do recognize that ROUGE and BLEU provide information on how similar a model's response is to the high-quality 'gold response'. We re-evaluated model generations on the FaithDial dataset on BLEU, ROUGE-1, and ROUGE-L, with the same nucleus + top-k sampling decoding method as in Table 1. Results are as follows:
> | Model              | BLEU | | | ROUGE-1 | | | ROUGE-L | | |
> | :---------------- | :------: | :------: | :------: | ----: |:------: | :------: | :------: | :------: | :------: |
> |   |   **Baseline**   | **+P-CRR** | **+ S-CRR** |    **Baseline**   | **+P-CRR** | **+ S-CRR** |   **Baseline**   | **+P-CRR** | **+ S-CRR** |
> | GPT2-Small           |  7.50 | **10.47** | 8.52 | 33.01 | **37.77** | 34.70 | 27.74 | **32.01** | 29.27 |
> | GPT2-Medium           |   8.03 | **10.83** | 9.00 | 33.93 | **39.08** | 36.22 | 28.28 | **33.32**  | 30.15 |
> | T5-Base          |   6.58 | **9.82** | 6.12 | 33.34 | **37.38** | 32.62 | 26.12 | **30.90**  | 25.38 |
> | OpenLlama-3B    |   8.54 | **11.11** | 9.04 | 35.85 | **39.69** | 37.27 | 29.65 | **33.39**  | 30.79 |
>
> * We can observe that **both P-CRR and S-CRR brings considerable performance gains** on BLEU and ROUGE metrics for all 4 models, further proving the effectiveness of our proposed method.
>   * In the final version, we will incorporate this ablation study result in the Experiment section.

---

> > ### Author Response · Authors · 2023-11-21
> > **Referenced Work**
> >
> > [1] Nallapati et al. Abstractive Text Summarization Using Sequence-to-Sequence RNNs and Beyond. CONLL 2016.
> >
> > [2] Ziwei Ji et al. Survey of Hallucination in Natural Language Generation. ACM Computing Surveys, Volume 55, Issue 12.
> >
> > [3] Chunting Zhou et al. Detecting hallucinated content in conditional neural sequence generation. ACL-IJCNLP 2021.
> >
> > [4] Bhuwan Dhingra et al. Handling divergent reference texts when evaluating table-to-text generation. ACL 2019.
> >
> > [5]  Esin Durmus et al. FEQA: A question answering evaluation framework for faithfulness assessment in abstractive summarization. ACL 2020.
> >
> > [6] Or Honovich et al. Q2: Evaluating factual consistency in knowledge-grounded dialogues via question generation and question answering. EMNLP 2021.
> >
> > [7] Kalpesh Krishna et al. Hurdles to progress in long-form question answering. NAACL-HLT 2021.

---

> > ### Comment · Reviewer_drdo · 2023-11-22
> >
> > Yep, you should provide such common metrics to show your performance.
> >
> > According to the provided new results, the improvement is `too impressive.`  For example, GPT-Small+P-CRR can significantly surpass OpenLlama-3B. Consequently, such results can not convince me.
> >
> > Empirically, your approach does not involve new data or new input; it is hard to guess why it can improve so much.  I am sorry, now I am doubting whether there are some tricks.

---

> > > ### Author Response · Authors · 2023-11-22
> > > **Response to Additional Concern of Reviewer**
> > >
> > > ### **[Q2 - Improvements are too impressive. There might be some tricks? ---- We DEEPLY VALUE research integrity, and we can ensure that NO tricks were used. Second, Since ROUGE and BLEU compare model responses to ground truth answers, it is possible that models like OpenLlama with more diverse vocabulary and longer generations could obtain a slightly lower score. ]**
> > >
> > > * We regret that the reviewer has the perception that tricks might be used for our evaluation results. As part of the NLP research community, we deeply value honesty and integrity in our research works, and we can ensure the reviewer that **NO TRICKS WERE USED FOR OUR EVALUATION**. We hereby provide an anonymized [Colab notebook](https://bit.ly/46sBtsw) with our evaluation pipeline and re-examined results for GPT2-Small, GPT2-Medium, T5-Base, and OpenLlama. Upon decision, **we will release our code and data** to ensure transparency.
> > > * For additional results, we would like to point out that although we mentioned how previous studies revealed the incapability of ROUGE and BLEU to reflect hallucination levels, **both P-CRR and S-CRR bring observable gains**  compared with the original decoding method without CRR.
> > > * Regarding the higher BLEU and ROUGE scores of smaller models, we would like to highlight the fact that in this task, overlap-based metrics such as ROUGE and BLEU provide information on **how similar a model's response is to the gold response**. Since the **gold responses in the FaithDial dataset tend to have less diverse vocabulary and shorter lengths**, it is possible that larger models (like Openllama) with longer answers, better generation ability, and more diversity in vocabulary and sentence structures get slightly lower overlap-based metrics.
> > >   * We also want to highlight the fact that the goal of the KGDG model is **NOT to generate responses that are similar to the gold response**, but to learn to leverage the provided knowledge and **generate faithful answers**. Therefore, we believe that the reported **Faithfulness score** also provides valuable information, for which **OpenLlama demonstrates the best performance in terms of faithfulness**.
> > > * We here also provide a brief qualitative analysis to validate our point that OpenLlama's responses are longer and have more diverse vocabulary, therefore **could be less similar to the gold response when judged with overlap-based metrics**. In this table, we compare the responses of GPT2-Small with P-CRR and that of the original OpenLlama.
> > >
> > > | Knowledge | Gold Response   | GPT2-Small + P-CRR | OpenLlama |
> > > | -------- | ------- | ------- | ------- |
> > > | Dylan's Candy Bar is **a chain of boutique candy shops and candy supplier currently located in New York City**; East Hampton, New York; Los Angeles, Chicago and Miami Beach, as well as in wholesale venues around the globe. | I don't know how good they are, but Dylan's Candy Bar has **a chain of candy shops** in various cities.  |  Dylan's Candy Bar is **a chain of candy shops and candy supplier currently located in New York City**.  | I don't know about a good brand, but I know about Dylan's Candy Bar, which is in Manhattan, East Hampton, LA, Chicago and Miami Beach as well. |
> > > | A number of **herbal concoctions and tonics have been used by strong men and athletes since ancient times** across cultures to try to increase their strength and stamina.  | Do you eat meat to increase your strength and stamina ? Athletes have used a number of **tonics and concoctions** to try to increase stamina and strength | I see, I don't know about the meats, but some **herbal concoctions and tonics have been used by strong men and athletes since ancient times**. | Did you know people used herbal concoctions for a variety of purposes? They have been used in ancient times to try to increase the strength of those athletes and since ancient times to try to help the stamina. |
> > > | It is evoked by light which **has a dominant wavelength of roughly 495570 nm**. | **The dominant wavelength of** green is around **495570 nm**.    | Cool!  It is evoked by light that **has a dominant wavelength of about 495570 nm**.   | Interesting, did you know that it is also evoked by light? That has a wavelength of 495570 nm. |
> > >
> > > * It can be observed from the above 3 examples that **GPT2-Small tends to quote part of the provided knowledge**, while **OpenLlama tends to paraphrase the information**, outputting longer answers with more diverse language structure and vocabulary. This is because the smaller model is not comparable with OpenLlama in terms of generation ability. Since the gold answers of FaithDial are also **shorter and overlap with the provided knowledge**, overlap-based metrics might slightly favor responses of GPT2-Small. However, we would like to **re-emphasize** that ROUGE and BLEU only measure some aspect of responses - namely how similar they are to the gold responses - and therefore conclusions on model performances could not be directly drawn from overlap-based results.

---

> ### Comment · Reviewer_drdo · 2023-11-22
>
> This answer do not directly answer my question. Besides,  as we know, BLEU & ROUGE cannot evaluate the diversity.

---

> ### Author Response · Authors · 2023-11-23
> **Response to Additional Concern of Reviewer**
>
> ### **[Q1 - Helpfulness of Model Responses Due to Low Diversity ---- We conducted string matching to find answers with "don't know" in the generated responses, and did not observe a significant increase in the percentage of such answers in CRR responses.]**
>
> * Considering the reviewer's original concern about decoding methods with CRR would result in outputting boring answers such as **"I don't know"**, we conduct string matching to find out the **percentages of responses** with the phrases "don't know", "don' t know", "don ' t know", or "don 't know" (to take into account formatting differences). Results are below:
> | Model              | Baseline | P-CRR | S-CRR |
> | :---------------- | :------ | :---- | :---- |
> | GPT2-Small        |   3.73%   |  3.98% | 3.53% |
> | GPT2-Medium         |   3.00%  | 3.56% | 2.88% |
> | T5-Base    |  8.41%   | 6.05% | 10.54% |
> | OpenLlama-3B |  5.07%   | 5.54% | 5.45% |
>
> * We can observe that when adopting P-CRR or S-CRR, there is **no significant increase** in the percentage of answers with the phrase "don't know". Should the reviewer have further questions or concerns about the helpfulness of model responses with CRR, we will provide further clarifications and results if necessary.
>
> ### **We would like to express our sincere gratitude to the reviewer for providing additional thoughts on our results. Please let us know if there are any further questions.**

---

### Official Review · Reviewer_Rt2T · 2023-10-30

**Soundness:** 2 fair
**Presentation:** 2 fair
**Contribution:** 2 fair
**Rating:** 3
**Confidence:** 4

**Summary:**

This paper presents sequence-level certainty as a common theme over hallucination. The authors categorize sequence-level certainty into probabilistic certainty and semantic certainty and explore the correlation between sequence-level certainty and the level of hallucination. Based on the observation, the authors proposed a decoding-time method called Certainty-based Response Ranking (CRR) to mitigate hallucination. The experimental results show CRR can reduce model hallucination under certain settings.

**Strengths:**

- The author's proposal of sequence-level certainty as a common theme over hallucination is a good idea, as is the design of probabilistic certainty and semantic certainty.
- The logic and structure of the paper is clear and easy to follow. The design details and experimental setup are explained thoroughly. This work is comprehensive, well-organized, and complete.

**Weaknesses:**

-	The last sentence of the first paragraph in the Introduction sets too narrow of a definition for hallucination in the KGDG task. In the current era of large models, we need to view the hallucination problem from a broader perspective, such as fact-conflicting and context-conflicting hallucinations [1], rather than limiting to input-conflicting hallucination. Meanwhile, judging from Table 6 in the Appendix, the cases are too simple and the chosen models like GPT-2 have insufficient capabilities. Through prompting, I found that gpt-3.5 does not hallucinate on those examples.
- Evaluating large models' hallucination phenomena at the sequence level using semantic certainty is an good idea, but the evaluation method for semantic certainty seems a bit crude. Also, the proposed decoding method simply selects better generations from the candidate set based on different metrics, which is quite similar to similar to Minimum Bayes Risk Decoding[2].
- The paper uses a RoBERTa-Large-based hallucination classification to evaluate whether generated text contains hallucination. However, the accuracy and effectiveness of this method for judging hallucination are not explained. Accurately assessing whether generated text hallucinates is fundamental to the analysis and experiments in this paper, so it is an important part.
- Hypothesis 2 proposed in Section 3.4 is an important basis for subsequent work, but there may be issues with the verification process. On one hand, it is reasonable that low certainty can lead to hallucination. But on the other hand, when models hallucinate, certainty is not necessarily low (especially under the consistency-based certainty evaluation designed by the authors). Based on my personal experience and experiments with LLM, they sometimes hallucinate confidently, i.e. sampling multiple times yields consistent outputs, especially for knowledge and numeric hallucinations (but I have not statistically verified this phenomenon rigorously). Also, the PBCC test is sensitive to class distribution. With imbalanced categories like fewer hallucination examples, it can still give a positively correlated relationship between certainty and hallucination, ignoring cases of high certainty hallucination. As described in 3.3.1 and Table 2, there is indeed a class imbalance in the FaithDial dataset under nucleus sampling, where hallucination examples make up less than 10% of the data.
- The analysis of efficiency for the P-CRR and S-CRR methods is missing. These multi-sampling approaches may greatly increase time and computational costs.
- The abstract contains too many unnecessary details and is somewhat convoluted. modifications are needed. Less essential details like the introduction of P-CRR and S-CRR can be briefly summarized. More explanation of the motivation behind this work could be added.

[1] Siren's Song in the AI Ocean: A Survey on Hallucination in Large Language Models

[2] Understanding the Properties of Minimum Bayes Risk Decoding in Neural Machine Translation

**Questions:**

Refer to weakness

---

> ### Author Response · Authors · 2023-11-21
> **Thanks & Response to Reviewer (1/7)**
>
> We thank the reviewer for recognizing the novelty in our method design and paper structure. Below, we address your questions.
> ### **[Q1 - The definition for hallucination in the KGDG task is too narrow. ---- We follow previous works' definition of hallucination on KGDG task, further exploration of factual or context-conflicting hallucinations on this task would be an interesting direction for future works.]**
>
> Previous work on KGDG [1] provided the following definition:
> > * Factually Consistent Response is **consistent with the provided knowledge**.
> > * Hallucinated Response is **not consistent with the knowledge** but **may still be correct**.
>
> An explanation to this definition would be because of the formulation of the KGDG task, for which the model should generate an answer **according to a specific piece of knowledge**. Therefore, it is most crucial to investigate whether the model generation is not factually consistent with the provided knowledge in the input. In our work, we follow the related previous works [1] [2] [3] to define hallucinations in this task as containing information that is unattributable to the source knowledge provided to the model.

---

> ### Author Response · Authors · 2023-11-21
> **Response to Reviewer (2/7)**
>
> ### **[Q2 - Cases are too simple and the chosen models like GPT-2 have insufficient capabilities. ChatGPT maybe would not hallucinate on these examples. ---- We included experiments on OpenLlama-3B,  a representative of the recent family of LLMs, with positive results. We did not experiment on ChatGPT because we have no access to model output distribution, and we cannot fine-tune it as we did for other models.]**
>
> * In order to explore the effectiveness and generalizability of our method to LLMs, we include full experiments with **OpenLlama**, a representative of the recently developed LLMs. By using Llama, we could **avoid black box scenarios and have full access to model output distribution**. We were also **able to fine-tune the model specifically for KGDG task**, aligning experimenting settings with other models. Positive results on OpenLlama provide empirical insights that **our proposed methods are generalizable to larger LLMs**, in addition to GPT-2 and T5.
>
> * We did not choose to experiment with ChatGPT for two major reasons:
>   * We **cannot obtain the output distribution**, therefore would not be able to experiment with methods related to probabilistic certainty.
>   * We **cannot fine-tuned ChatGPT for KGDG task**, so only prompt-based methods could be used to sample the answers. Since all the reported experimented models are fine-tuned on the KGDG task, experimenting with ChatGPT through prompting would **cause the experiment setting to be very different from the other models**.

---

> ### Author Response · Authors · 2023-11-21
> **Response to Reviewer (3/7)**
>
> ### **[Q3 - Semantic certainty evaluation method seems crude. The method also appears similar to Minimum Bayes Risk Decoding. ---- Our simple yet effective method lays great foundation for future works. While the method setting that selects from a candidate set of generations is a common trait among multi-sampling-based methods, we believe that CRR is inherently different from MBR Decoding. ]**
>
> * We believe that the **simple yet effective design** of our proposed CRR methods lays good foundations for future work, which might make improvements on top of our proposed approach.
>   * Regarding your question on the similarity between our CRR approach and the Minimum Bayes Risk Decoding approach, we believe that **CRR is inherently different from MBR Decoding**. Having its root in decision theory, MBR Decoding has recently been applied to tasks such as Automatic Speech Recognition (ASR), and Neural Machin Translation (NMT), with **no related exploration on how they can be applied to address model hallucination**. For the NMT task, previous work [4] have stated that:
> > The goal of MBR is to find **not the most probable translation**, but the one that minimizes the expected risk for a given loss function and the true posterior distribution.
>
>   * Therefore, **MBR Decoding diverges from the definition of our P-CRR approach**, which intuitively outputs the candidate with largest sequence-level probabilistic certainty.
>   * Although the *sample>compare>choose* pipeline might appear similar to S-CRR, we believe that **this similarity is shared as a method structure by a family of multi-sampling-based generation methods** [5][6][7], each independently designed **with variations** to address different domain-specific tasks.
>   * In addition, the utility function for MBR Decoding utilizes **overlap-based metrics** such as ROUGE and BLEU to calculate the token or n-gram component-level similarity of sampled responses. Our S-CRR method considers **semantic similarity or entailment** when choosing from generation candidates, rather than component or token-level similarity.

---

> ### Author Response · Authors · 2023-11-21
> **Response to Reviewer (4/7)**
>
> ### **[Q4 - Accuracy and effectiveness of the hallucination critic model are not provided. ---- We utilized an off-the-shelf hallucination classification model, and are happy to quote the original authors' reported metrics to prove its effectiveness.]**
>
> * For hallucination classification, we use an off-the-shelf hallucination classification model that is specifically tuned for recognizing hallucinations on the KGDG task. It was trained on FaithCritic, a derivative of the FaithDial dataset [2]. The objective of the model is to predict whether an utterance is faithful or not, given the source knowledge.
> * We hereby quote the reported accuracy of the model provided by the original authors, along with the ablation experiment results of models trained on 3 other NLI datasets (DECODE, DNLI, MNLI) and inference on 2 hallucination classification datasets (BEGIN, FaithCritic):
> | Trained on              | BEGIN | FaithCritic |
> | :---------------- | :------ | :----|
> | DECODE        |   58.8   | 38.5 |
> | DNLI           |   59.8   | 30.9 |
> | MNLI    |  61.1   | 81.6 |
> | **FaithCritic** |  **71.6**   | **86.5** |
> *  We can observe that the **best hallucination detection performance is achieved by the FaithCritic model**.
>    * Although the FaithCritic model might not be perfect, **it is among the best-performing hallucination classification models available**, and therefore we choose to report results using this model.

---

> ### Author Response · Authors · 2023-11-21
> **Response to Reviewer (5/7)**
>
> ### **[Q5 - Models can sometimes hallucinate confidently. PBCC test might be sensitive to class distribution in the data. ---- Although edge cases do exist, our experiments show a general correlation between high sequence-level certainty and low hallucination. We provide the mean differences in sequence-level certainty scores between hallucinated and faithful answers to validate PBCC results.]**
> * We agree that there exists edge cases where a generative model hallucinates with high confidence. However, we believe that **our experiments empirically show a general correlation** between low sequence-level certainty and high hallucination on KGDG task.
> * In order to validate PBCC test results that the sequence-level certainty level of faithful answers are higher than that of hallucinated answers, we hereby report the **mean differences** in the sequence-level certainty scores **between faithful and hallucinated answers**, on the FaithDial dataset, with the same nucleus + top-k sampling decoding method as in Table 1.
>   * Specifically, we first group model responses into $R_{Faithful} = r_{f,1}, r_{f, 2}, ... r_{f, m}$ and $R_{Hallucinated} = r_{h, 1}, r_{h, 2}, ... r_{h, n}$.
>   * Then, let $Certainty(\cdot)$ be a function that calculated the sequence-level certainty score of a response. We report the mean differences between the certainty scores of these two groups of responses as:
> $$
> \text{Mean Diff.} = \frac{1}{m} \sum_{i=1}^m Certainty(r_{f, i}) - \frac{1}{n} \sum_{j=1}^n Certainty(r_{h, j})
> $$
> * Results are as follows:
> | Model              | P-CRR Mean Diff. | S-CRR Mean Diff. |
> | :---------------- | :------: | ----: |
> | GPT2-Small        |   0.17   | 0.63 |
> | GPT2-Medium         |   0.20   | 0.40 |
> | T5-Base    |  0.14   | 0.08 |
> | OpenLlama-3B |  0.04   | 0.54 |
> * The **positive mean differences** between certainty scores of faithful responses and hallucinated responses for **all 4 models** further validates our PBCC test results, indicating that **faithful responses have significantly higher sequence-level certainty scores** - both in terms of semantic certainty and probabilistic certainty - than hallucinated responses.

---

> ### Author Response · Authors · 2023-11-21
> **Response to Reviewer (6/7)**
>
> ### **[Q6 - No analysis of efficiency for the P-CRR and S-CRR methods. ---- P-CRR does not introduce additional inference time, as the candidate sampling process is simultaneous. S-CRR introduces additional inference costs, but can be applied to black-box scenarios.]**
> * P-CRR method **does not introduce additional inference time** as it is probability-based, and the multi-candidate sampling is done simultaneously. However, model output probabilities must be available for implementing the P-CRR approach.
>   * This is one of the motivations of us to study semantic certainty and S-CRR, because it can be implemented even in blackbox scenarios where only textual outputs are available.
> * S-CRR does introduce additional inference costs in the Agreement Score calculation process. However, since we compute semantic certainty on sequence level and the Roberta-Large model we use for Agreement Score calculation does not have a tremendous size (355M), the inference time isn't significantly affected (the inference time is ~1.5 times that of baseline when number of sampled candidates is set to 5). In addition, S-CRR is **a promising approach for application in black-box scenarios**, since it **does not require access to output probabilities**.
>   * In the final version, we will rerun experiments and report inference time for all methods in the Appendix.
> * Strong empirical results on P-CRR and S-CRR prove the **correctness** and **effectiveness** of the proposed **sequence-level certainty** method for **measuring and potentially reducing hallucinations**.

---

> ### Author Response · Authors · 2023-11-21
> **Response to Reviewer (7/7)**
>
> ### **[Q7 - The abstract can be more concise. --- Thanks for pointing out! We will make modifications in the final version.]**
>
> We thank the reviewer for comments regarding the presentation of the abstract. We will make further edits in the final version to make it more concise, as well as add in necessary technical details.

---

> > ### Author Response · Authors · 2023-11-21
> > **Referenced Work**
> >
> > [1] Sashank Santhanam et al. Rome was built in 1776: A Case Study on Factual Correctness in Knowledge-Grounded Response Generation. CoRR 2021.
> >
> > [2] Nouha Dziri et al. FAITHDIAL: A Faithful Benchmark for Information-Seeking Dialogue. TACL 2022.
> >
> > [3] Hannah Rashkin et al. Increasing Faithfulness in Knowledge-Grounded Dialogue with Controllable Features. ACL 2021.
> >
> > [4] Mathias Muller and Rico Sennrich. Understanding the Properties of Minimum Bayes Risk Decoding in Neural Machine Translation. ACL 2021.
> >
> > [5] Xuezhi Wang et al. SELF-CONSISTENCY IMPROVES CHAIN OF THOUGHT REASONING IN LANGUAGE MODELS. ICLR 2023.
> >
> > [6] Mingxing Duan et al. A Novel Multi-Sample Generation Method for Adversarial Attacks. ACM Transactions on Multimedia Computing, Communications, and Applications, Volume 18, Issue 4.
> >
> > [7] Erci Mitchell et al. Enhancing Self-Consistency and Performance of Pre-Trained Language Models through Natural Language Inference. EMNLP 2022.